# Dual-Functional Iron Oxide Nanoparticles Coated with Polyvinyl Alcohol/5-Fluorouracil/Zinc-Aluminium-Layered Double Hydroxide for a Simultaneous Drug and Target Delivery System

**DOI:** 10.3390/polym13060855

**Published:** 2021-03-10

**Authors:** Mona Ebadi, Saifullah Bullo, Kalaivani Buskaran, Mohd Zobir Hussein, Sharida Fakurazi, Giorgia Pastorin

**Affiliations:** 1Materials Synthesis and Characterization Laboratory, Institute of Advanced Technology (ITMA), Universiti Putra Malaysia, Serdang 43400, Malaysia; mona.ebadi64@gmail.com (M.E.); bullosaif1@gmail.com (S.B.); 2Department of Management Sciences and Technology, Begum Nusrat Bhutto Women, University Sukkur, Sindh 65200, Pakistan; 3Laboratory of Vaccine and Immunotherapeutics, Institute of Bioscience, Universiti Putra Malaysia, Serdang 43400, Malaysia; vaneey88@yahoo.com (K.B.); sharida.fakurazi@gmail.com (S.F.); 4Department of Human Anatomy, Faculty of Medicine and Health Sciences, Universiti Putra Malaysia, Serdang 43400, Malaysia; 5Department of Pharmacy, National University of Singapore, Singapore 117543, Singapore; phapg@nus.edu.sg

**Keywords:** nanoparticles, Fe_3_O_4_ nanoparticles, layered double hydroxide, polymeric coating

## Abstract

Iron oxide nanoparticles are suitable for biomedical applications owing to their ability to anchor to various active agents and drugs, unique magnetic properties, nontoxicity, and biocompatibility. In this work, the physico-chemical and magnetic properties, as well as the cytotoxicity, of Fe_3_O_4_ nanoparticles coated with a polymeric carrier and loaded with a 5-fluorouracil (5-FU) anti-cancer drug are discussed. The synthesized Fe_3_O_4_ nanoparticles were coated with polyvinyl alcohol and Zn/Al-layered double hydroxide as the drug host. The XRD, DTA/TG, and FTIR analyzes confirmed the presence of the coating layer on the surface of nanoparticles. The results showed a decrease in saturation magnetization of bare Fe_3_O_4_ nanoparticles after coating with the PVA/5FU/Zn/Al-LDH layer. In addition, the presence of the coating prevented the agglomeration of nanoparticles. Furthermore, the pseudo-second-order equation governed the kinetics of drug release. Finally, the coated nanoparticles showed stronger activity against liver cancer cells (HepG2) compared to that of the naked 5-FU drug, and displayed no cytotoxicity towards 3T3 fibroblast cell lines. The results of the present study demonstrate the potential of a nano delivery system for cancer treatment.

## 1. Introduction

In recent years, the application of new materials and the efficiency of the products have been considerably improved using nanomaterial technology [1]. The reason for this is that the properties of nanomaterials dramatically change and become unique in a nano-sized regime. These nanoparticles are useful for various applications in drug delivery, clinical and pharmaceutical industries, and nanomedicine.

Cancer is one of the most lethal diseases [2,3,4]. Treatment methods such as chemotherapy can be potentially useful for cancer treatment, but unfortunately, non-cancerous tissues and other parts of the body are also usually affected [5]. For this reason, researchers have exploited nanotechnology for cancer treatment. This novel method is superior compared to conventional methods. The use and application of magnetic nanoparticles and more specifically, targeted drug delivery achieved by anchoring the drug to magnetic nanoparticles, are among the strategies that can be used [6,7]. These magnetic nanomaterials can be used for drug delivery, cellular separation, and cancer treatment [8,9]. The magnetic nanoparticles also act as carriers to transport the drugs [10].

A multifunctional drug delivery system plays a significant role in carrying a variety of drugs and other agents to a specific target of the human body with sustained- or controlled-release [11,12]. With this system, there is a possibility to achieve a stable and uniform presence of drugs for a certain period, and fluctuations of the drug concentration over and below the therapeutic window can be eliminated. Therefore, a maximum level of the therapeutic window of the drug will reach the tissues, leading to fewer adverse side effects and an increase in the drug efficacy. An ideal drug delivery system for treatment should have excellent biodegradability, biocompatibility, and convenience for the patients during and after drug administration [13]. Furthermore, a high drug loading capability is desired and the synthesis process should be simple and cost-effective [8].

Due to the therapeutic role of the drug, various drug delivery systems, such as targeted drug delivery systems, have been developed and investigated to prevent drug degradation at the desired part of the body, and protect it until the drug reaches the targeted tissue or tumor cell area, maintaining its biological properties. Some of the drugs are extremely toxic and can damage healthy cells and cause side effects, and their degradation during release can decrease their therapeutic effects [14]. In general, compared to normal cells, cancer cells display abnormal cell proliferation via dysregulation of the progression cycle. The mechanism by which an anticancer drug kills the cancer cells occurs via the arrest of cell growth and prevention of the cell cycle. Because dividing cells are rapidly affected by the anticancer drug, normal cells are also affected.

Therefore, the use of carriers in drug delivery systems is of great importance. Generally, the reason for using these carriers in drug delivery systems is to achieve systems with suitable drug loading and release properties with no or a low toxicity and high half-lives of the drugs. In addition, carriers should deliver the maximum amount of drug to the desired place [15,16].

Various anti-cancer drug carriers, such as magnetic iron oxide nanoparticles, layered double hydroxide, and synthetic and natural biodegradable or non-biodegradable polymers, have been used in targeted drug delivery systems, where the drugs can be either adsorbed on the surface of these carriers or encapsulated inside them. If magnetic nanoparticles are used as cores and are coated with polymeric shells that contain the drug, then they can function as an agent to guide the drug to a specific place using an external magnetic as the guide. In addition, some of the polymeric materials can respond and lead to drug delivery, depending on the surrounding environment [17,18,19,20].

Controlled drug release is the process in which a suitable carrier is carefully mixed with a drug or active agent and the drug is released in a desired and predetermined manner. Controlled drug release is a major factor of targeted drug delivery. The potential mechanisms of drug release are (a) return and release of the adsorbed drug on the surface, (b) infiltration from its carrier matrix, (c) infiltration from the carrier wall for micro- and nano-particles and nanocapsules, and (d) erosion and destruction of the carrier matrix [21,22].

Although polymeric nanoparticles have been used as suitable drug delivery agents, there are not many reports available to describe their use for the construction of a multi-functional theranostics delivery agent, where two or more active agents are loaded simultaneously. In this work, the synthesis of multi-functional theranostics, magnetic iron oxide nanoparticles by the co-precipitation method is described. These magnetic nanoparticles were coated by a drug carrier and then subsequently loaded with the drug. Finally, cell cytotoxicity studies were performed on HepG2 liver cancer cells and 3T3 fibroblast cells to assess their biocompatibility.

## 2. Materials and Methods

### 2.1. Materials

Iron (III) chloride (99%, molecular weight 162.20 Mwt), iron (II) chloride (99%, Molecular weight 126.20 Mwt), and ammonia 25% were purchased from Merck (Kenilworth, NJ, USA) with laboratory purity. Polyvinyl alcohol was obtained from Acros Organics (PVA, Fair Lawn, NJ, USA, average M.W. 6000). For the synthesis of Zn/Al-LDH, aqueous zinc nitrate six-hydrated, aqueous aluminum nitrate nine-hydrated, and sodium hydroxide (99%, ChemAR, Kielce, Poland) were used. 5-fluorouracil (5-FU) (an anti-cancer drug, C_4_H_3_FN_2_O_2_, 98%)) from AKSci (Union City, CA, USA) and dimethylsulfoxide ((CH_3_)_2_SO, DMSO) were purchased from Sigma Aldrich (St. Louis, MO, USA). All chemicals were used with laboratory purity and without previous purification. Deionized water was used in all stages of the experiment.

### 2.2. The Provision of Samples

The co-precipitation method was used to synthesize Fe_3_O_4_ nanoparticles [15]. The reaction process (1) by which Fe_3_O_4_ nanoparticles are formed from two soluble salts is as follows:2FeCl_3_·6H_2_O + FeCl_2_·4H_2_O → Fe_3_O_4_ + 8NH_4_Cl + 20H_2_O(1)

FeCl_3_·6H_2_O (0.99 g) and FeCl_2_·4H_2_O (2.4 g) were separately prepared in distilled water with a ratio of 2:1 and mixed in a glass multi-wall reactor. After a few minutes, 6 mL ammonia was immediately added to the reactor and stirred with a magnetic stirrer for 15 min. In this stage, a black precipitate formed, indicating the formation of Fe_3_O_4_. Then, the sample was washed three times. After that, 2% of polyvinyl alcohol was dissolved in the black precipitate by placing it in an autoclave at 150 °C for one day. The uncoated nanoparticles were eliminated by washing them using deionized water (DI) water and dried at 70 °C for 8 h. Typically, a base solution (5-FU dissolved in dimethyl sulfoxide) was added to the prepared black powder under the magnetic mixer for the whole day. To prepare the Zn/Al-LDH using the co-precipitation method, at first, a solution of Al(NO_3_)_3_·9H_2_O and Zn(NO_3_)_2_·6H_2_O salts with a molar ratio of Zn/Al = 4 in a four-mouth flask under a nitrogen gas environment was prepared. The NaOH solution was added into the solution by dropwise addition at a rate of 1 drop per minute. The adding process of NaOH was continued until a solution with a pH of 7 was reached. Then, 3 g of prepared nanoparticles was mixed with Zn/Al-LDH. Upon completion, sample sediment was collected by centrifugation, washed three times, and dried in an oven.

### 2.3. Characterization

The XRD analysis was performed to determine the type of phase present in the sample. A Shimadzu XRD PW-6000 Model (Kyoto, Japan) using monochromatic CuK𝛼 radiation (𝜆 = 1.5406 Å) at 40 kV-30 mA in the range of 2–80° was used. To determine the surface coating entity of the nanoparticles, Fourier transformed infrared spectroscopy (FTIR) was used. The analysis was performed using a Thermo Nicolet 6700 (the Auxiliary Experiment Module (AEM), Madison, WI, USA) device at a resolution of 0.09 cm^−1^ in the range of 500 to 4000 cm^−1^. Samples were assessed using the potassium bromide (KBr) disk method. To investigate the thermal behavior and change in the amounts of the drug, polymer, Fe_3_O_4_ nanoparticles, and layered double hydroxides (LDH) weight loss, thermogravimetric analysis and differential thermogravimetric analysis were conducted to identify the mass reduction using a Mettler–Toledo model (TGA/DTG, Greifensee, Switzerland) in the range of 20 to 1000 °C.

All the samples were tested in a powder form. Pure iron oxide nanoparticle, polymer, drug, and LDH samples were also evaluated as reference samples. The average particle size and the particle size distribution plot were examined via dynamic light scattering, using a MALVERN (NanoS, Malvern, UK) device.

The surface morphology and elemental compositions of the particles were recorded using a NOVA NANOSEM 230 model field emission-scanning electron microscope (Los angeles, CA, USA) and energy-dispersive X-ray spectroscopy, respectively. Energy-dispersive X-ray spectroscopy (EDX) was used to probe and determine the chemical composition and identify the elements that were present in the samples.

For understanding the magnetic properties of the nanoparticles, a vibrating sample magnetometer device—model Lakeshore 4704 of the Lakeshore cryotronics company (Westerville, Ohio, United States of America)—measured in the range of −10,000 to 10,000 Oe was used. The morphology and size of the nanoparticles of the synthesized samples were evaluated using high-resolution scanning transmission electron microscopy (HRTEM, Hitachi H-7100, Tokyo, Japan). The Image J software was used to obtain the particle size distribution of the sample by choosing more than 100 particles. The drug loading and drug release behavior were then measured by a UV-visible spectrophotometer using a Perkin Elmer, Lambda 35, UV-Visible spectrometer at λmax = 214 nm. Inductively coupled plasma-optical emission spectrometry (Optima 8300, Perkin Elmer, Wellesley, MA, USA) was employed to identify the elements Mg, Al, and Fe and their concentrations. A CHN analyzer (LECO, TruSpec, Stockport, UK) was used to measure the percentage of N, H, and C elements. The magnetic characteristic was investigated using a vibrating sample magnetometer analysis in a measuring range of 20,000 to 20,000 G.

For evaluation of cellular toxicity and cell viability assay, normal human fibroblast (3T3) and human hepatocellular carcinoma cells (HepG2) that were obtained from American Type Culture Collection (ATCC) (Manassas, VA, USA) were used. The cells were cultured in Roswell Park Memorial Institute, 1640 medium (Nacalai Tesque, Kyoto, Japan) containing 10% fetal bovine albumin (Sigma-Aldrich, St. Louis, MO, USA) along with 1% antibiotics (included 10,000 units/mL penicillin/10,000 μg/mL streptomycin0 (Nacalai Tesque, Kyoto, Japan) in A T75 flask in a 37 °C 5% CO_2_ incubator. When the cell layers were harvested via 0.25% trypsin/1mM-EDTA (Nacalai Tesque, Kyoto, Japan), they were seeded in a plate of a 96-well tissue culture with 1.0 × 10^4^ cells per well for one day to attach and about 80% confluence was attained for treatment. After one day of attachment to the respective wells, cells were treated with specific values of the treatment sample (1.25 to 100 μg/mL) consisting of Fe_3_O_4_ nanoparticles-polyvinyl alcohol (FPVA), Fe_3_O_4_ nanoparticles-polyvinyl alcohol-zinc/aluminum layered double hydroxide (FPVA-ZLDH), pristine 5-FU, and FPVA-5FU-ZLDH. The treatment solutions were prepared by dissolving the compound in 1% DMSO and RPMI at a ratio of 1:1 and then diluting it in the same media to produce various concentrations from 1.25 to 100 μg/mL. After incubation at 24 and 72 h, the viability of the cells and cytotoxicity were assessed by the methylthiazol tetrazolium (MTT)-based assay. For this purpose, 10 μL of MTT solution (5 mg/mL in PBS) was added to each well and placed in an incubator for 3 h before being aspirated. Then, purple formazan salt formed was dissolved in 100 μL DMSO solution in the dark and at room temperature. The absorption rate of the microplate reader device (Biotek LE800, Winooski, VT, USA), which reflects cell growth, was characterized at 570 nm. The intensity of the color is proportional to the number of cells that are metabolically active and therefore to the viability of the cells. For the computation of the half-maximal inhibitory concentration (IC_50_), the x-axis against the y-axis was the plot. Plotting the xy graph using the logarithmic value of their concentration and the following curve fitting (nonlinear regression) under the xy analysis to find a straight line equation fit (y = ax + b) was used to calculate the IC50 values. Several pieces of software were used for statistical analysis, including SPSS, ANOVA, and Duncan’s Multiple Range Test. The pristine 5-FU and FPVAFU-ZLDH (nanoparticles) were significantly different (* *p* value < 0.5).

## 3. Results and Discussion

### 3.1. X-ray Diffraction

The XRD results are depicted in Figure 1. The relatively wide peaks in the figure are related to the small particle size. Due to the great similarity between the XRD results of magnetite and maghemite, the distinction of these two phases is not straightforward. However, the formation of the maghemite phase is unlikely, owing to the fact that the synthesis was conducted under non-oxidative conditions [23]. Figure 1A displays the X-ray diffraction pattern of Fe3O4 nanoparticles produced by the co-precipitation method. The characteristic peaks in the XRD pattern showed peaks at the 2θ-diffraction angles of 35°, 41.3°, 50.4°, 62.9°, 67.2°, and 74.1°, which correspond to the crystalline planes of (220), (311), (400), (422), (511), and (440), respectively (JCPDS card numbers 19-629) [24].

Based on the XRD analysis, all the samples prepared after optimization were found to be of a pure phase. As can be observed from the XRD result, the final sample has the characteristic peaks of LDH, wherein the regions between the 2θ = 11°–14.5° and also 22° are ascribed to the (003) and (006) peaks, respectively. The XRD diffraction pattern of the sample after the polymer coating showed a peak of PVA at 2θ = 19.5°. Another sharp peak at 28.4° was due to the presence of 5-FU. The XRD peak intensity of the sample was enhanced with an increase of the polyvinyl alcohol and other coating agents, which indicated a better crystalline structure in the resulting sample [25,26]. The displacement of some peaks to higher angles, as well as an increment in the peak width, were apparent.

### 3.2. Fourier Transform Infrared Spectroscopy Spectra

The FTIR results of the synthesized nanoparticles and the pure Fe_3_O_4_ nanoparticle, polymer, drug, and Zn/Al-LDH samples are shown in Figure 2. As can be observed in Figure 2E, Fe_3_O_4_ nanoparticles show sharp bands at 537 cm^−1^ and in the range of 4000 to 500 cm^−1^. This band is related to the stretching bond of Fe−O in the tetrahedral and octahedral conditions of the magnetite phase. The absorbance bands at wavenumbers of 3340, 2917, 1750, and 1478 cm^−1^, as characteristic bands of polyvinyl alcohol, are related to the O−H stretching from the intermolecular and intramolecular hydrogen bonds, the C−H bending, the C=O and C−O stretching, and C−H2 bonds, respectively. The band at 1582 cm^−1^ is attributed to interlayer water adsorption [27]. It was found that the polymer was present on the surface of the sample. The bands at 1011 and 780 cm^−1^ are related to the C–O bonds and O–M–O lattice vibration, respectively. Moreover, the peak at 1228 cm^−1^ indicated the stretching vibration for the N−H bond.

In the case of samples containing nitrate, the band at around 1384 cm^−1^ is ascribed to the asymmetric stretching vibration (v3) of nitrate ions [28]. The obvious presence of this band revealed a nitrate substitution in the structure. The band shifts to the lower wavenumbers in this sample (1348 cm^−1^) revealed that the hydrocarbon chains are surrounded by the surface of the nanoparticles.

### 3.3. Thermal Analysis

Figure 3 shows the TGA/DTG result of MPVAFU-ZLDH and the data are tabulated in Table 1. Generally, there are three endothermic regions, as can be seen in the figure. The first and second weight loss occurring up until 160 °C is related to the evaporation and elimination of surface water molecules and an interlayer water molecule; no weight loss was observed below 50 °C. The third sharp weight loss that continues to 330 °C is ascribed to the reduction of nitrate to nitrite ions, along with the loss of surface-active agents. The fourth weight loss that occurred at 330−400 °C is attributed to the polymer and metal oxide decomposition (with a weight loss of 1.9%). The fifth endothermic peak continuing up to 600 °C (with weight loss of around 4.2%) is attributed to the removal of the majority of hydroxyl functional groups, together with the decomposition of nitrate, and the weight loss in this stage is due to the complete removal of NO2 from the sample. Heating the samples to above 800 °C led to the destruction of the layered structure of LDH and the formation of a new crystalline oxide phase [24,29].

The thermal gravimetric plot indicates complete decomposition of the sample until around 800 °C and the total weight loss was almost 30%. In the first region until 160 °C, the sample weight loss is 5.6% of its initial weight. In the second region, which continues to 330 °C, the weight loss is around 9.5%.

Three stages of ZLDH thermal decomposition could be expressed by the following formula.

In the first stage, dehydration occurs:Ca_1−x_Al_x_(OH)_2_(NO_3_)·nH_2_O → Ca_1−x_Al_x_(OH)_2_(NO_3_)_x_ + nH_2_O(2)

In the second and third stages, nitrate is transformed into nitrite and nitrate ions are removed from the system:Ca_1−x_Al_x_(OH)_2_(NO_3_)_x_·n → Ca_1−x_Al_x_O(NO_3_)_x_ + H_2_O(3)
Ca_1−x_Al_x_O(NO_3_)_x_ → Ca_1−x_Al_x_O_(1+x/2)_ + xNO_2_ + (x/4)O_2_(4)

### 3.4. Magnetic Properties Evaluation

The investigation of the magnetic characteristic of naked Fe_3_O_4_ and FPVAFU-ZLDH nanoparticles showed a hysteresis value close to zero with superparamagnetic properties (Figure 4). The saturation magnetization (Ms) of magnetic nanoparticles was found to be 27 emu/g according to the VSM analysis. Both the high value of saturation magnetization and superparamagnetic characteristics are among the important properties for the application of magnetic nanoparticles in the field of medical and drug delivery systems.

Topcaya et al. previously fabricated a magnetite nanoparticle coating with PVP and their investigation on the magnetic properties of the produced sample demonstrated a reduction in saturation magnetization in the coated Fe_3_O_4_ sample compared to that of non-coated Fe_3_O_4_. The decrease was attributed to the covering effects of PVP layers. Moreover, the presence of PVP diminished the magnetic dipolar interactions between the neighboring particles and as a consequence, the magnetism value was reduced [30]. The same observations can be seen in our synthesized samples with the presence of coating agents, such as polymers and layered double hydroxide. The results are listed in Table 2.

### 3.5. Surface Morphology

Figure 5 shows FESEM micrographs of naked Fe_3_O_4_ nanoparticles and the nanoparticles coated with PVA. As can be observed, the morphology of the particles was spherical and uniform, but with random aggregation. From the comparison of Figure 5A,B, it can be seen that the surface coating agent resulted in less agglomeration of the nanoparticles. Figure 5C shows the EDX spectra of FPVA-FU-ZLDH in terms of oxygen, carbon, aluminum, iron, and zinc. As expected, consistent with the EDX analysis of the FPVA-FU-ZLDH nanocomposite, the atomic weight percentage of iron, oxygen, carbon, zinc, and aluminum was estimated to be 18.1 25.9, 8.3, 2.9, and 44.7% wt, respectively. The presence of Fe, C, and O, in addition to Zn and Al elements, represents the formation of coating layers on the Fe_3_O_4_ surface.

The result of the DLS analysis (hydrodynamic size) of the synthesized samples is shown in Figure 6. As can be observed, the particle size distribution for the sample of FPVAFU-ZLDH is narrow and close to the normal distribution; the average particle size for the sample is smaller than the size of pure Fe_3_O_4_ nanoparticles. The hydrodynamic diameter for Fe_3_O_4_ and FPVA-FU-ZLDH nanoparticles is 295 and 122 nm, respectively, with a monomodal distribution for both of them. This indicates less agglomeration of the sample, which prevents particle size growth in the FPVAFU-ZLDH sample, and this can be attributed to the number of coatings, polymer, drug, and LDH. The reason for the presence of agglomerated particles in the final sample related to the fact that, during the co-precipitation method applied for the production of the sample, the particles are tightly attached to each other and form aggregates. Under this condition, an interaction is established between the surface anions and therefore, they attract each other, leading to the formation of aggregates.

### 3.6. HRTEM of the Sample

Figure 7 shows the HRTEM morphology and size of the uncoated Fe_3_O_4_ nanoparticles, which exhibit spherical shapes with a mean diameter of 34 nm and a monomodal distribution. The mean size was obtained by a calculation of more than 100 random particles through the ImageJ software. Based on the image, magnetic nanoparticles are inclined to agglomerate due to the bipolar–bipolar interactions and van der Waals forces between the magnetite nanoparticles [31,32,33]. In order to reduce the agglomeration, the magnetite nanoparticle surface was covered by layers of coating agents during production, reducing the agglomeration of nanoparticles and forming relatively uniform nanoparticles by creating repulsion among them. This resulted in the formation of a negative charge on the surface. Nevertheless, some of the drugs that should be placed in the interlayers may also be adsorbed on the surface of LDH. This causes the advent of surface forces, which leads to agglomeration and adhesion of the nanoparticles. This causes a reduction in the particle size with coating compared to that of the uncoated counterpart (Figure 7D).

### 3.7. Elemental Analyses

Elemental analyses are used to determine the chemical composition of a sample. In this study, the elemental content (Al, Fe, Mg, and Zn) and carbon, hydrogen, nitrogen, and sulfur were obtained by the ICP-OES and CHN methods. The results of the ICP-OES and CHN analyses are shown in Table 3. The elemental composition was 2.5%, 2.8%, and 24.8% for aluminum, zinc, and iron, respectively. This confirms the presence of ZLDH and a magnetic core in the sample. The presence of C, H, and N indicates the presence of polymer as the coating layer and the drug 5-FU. These results are in good agreement with the data obtained by the EDX analysis. The percentages of elements are summarized in Table 4.

### 3.8. In Vitro Release of 5-Fluorouracil from the Magnetic Nanoparticles

As can be observed in Figure 8, the percentage total release profiles of 5-FU from the synthesized nanoparticles was 94% in the solutions of phosphate-buffered saline at pH 4.8 and 91% at pH 7.4 as the release media within 7 days. These pH 4.8 and 7.4 buffer solutions were used to simulate the pH of blood (an extracellular lysosomal environment) and cancer cells (an intracellular lysosomal environment), respectively. The plot shows that in the first 33 h, the release of drugs was about 80%. However, no fast burst release can be observed. This amount of drug release could be due to drug adsorption on the surface of the nanoparticles and drug accumulation on the outer layers of the nanoparticles. After the fast initial release, the release behavior was controlled by anion replacement and the drug was released in a more controlled manner. Under this condition, the available anions in the buffer solution were exchanged with the interlayer drug anions through infiltration and anion replacement or ion exchange mechanisms. The bond between these anions and hydroxyl was strong after the replacement and was thus not replaced with other anions in the solution. As time proceeded, more replacement of the buffer anions took place, and therefore, the release of the drug from the nanoparticles became more difficult because the drug that was placed in the center or deeper into the layers had to cross a very complex pathway to pass out of the layers. Therefore, less drug release was observed.

The release of 5-FU profiles from the physical mixture was measured in two different pHs (7.4 and 4.8) of PBS solutions to mimic the environment in the stomach and body fluids, respectively. The release profiles are shown in Figure 9. Under both pH environments, no 100% release of the drug was observed and around 55% final release was found for both of them. Although a similar amount of release was observed, to achieve 50% release, the former only needed 7 min, compared to 13 min for the latter. The examination of the drug release into the two pHs indicated that the release was higher in the acidic solution compared to the slightly alkaline pH.

### 3.9. Release Kinetics of 5-Fluorouracil from the Magnetic Nanoparticles

The kinetics profiles for the release of 5-FU from its magnetic nanoparticles into the phosphate-buffered saline solutions at both pHs were analyzed using different kinetic models, including first-order (Equation (1)), pseudo-second-order (Equation (2)), and parabolic diffusion (Equation (3)). The data obtained from the three models were evaluated based on their correlation coefficient, rate constant, and half-life.
Pseudo-first order: ln (q_e_ − q_t_) = lnq_t_ − kt(5)
(6)$$\mathrm{Pseudo}-\mathrm{second}\mathrm{order}:t/q_{t}=1/kqe2+t/q_{e}$$
(7)$$\mathrm{Parabolic}\;\mathrm{diffusion}:(1-M_{t}/M_{o})/t=kt-0.5+b$$
where q_e_ is the quantity of drug release at equilibrium, q_t_ is the amount of drug release at a particular time t, k is the release constant, Mo is the drug remaining at release time 0, and M_t_ is the drug content remaining at release time t. The correlation coefficients could be deduced from the t/q_t_ versus t plot, as shown in Figure 10. The kinetics models could be used to investigate the kinetics of drug release, which included the linear regression, saturation release, rate constant, and t1/2, as given in Table 5.

The release of the 5-FU drug from the synthesized nanoparticles was found to obey the pseudo-second-order model. In this model, the values of the correlation coefficients, R^2^, were found to be %9987 and %9999 and the rate constants, K, were 4.31 × 10^−3^ and 4.33 × 10^−3^ mg/min at pH 7.4 and 4.8, respectively.

### 3.10. In Vitro Bioassay

#### 3.10.1. Cytotoxicity Studies on Normal 3T3 Fibroblast Cells

The cytotoxicity evaluation was carried out through treating Fe, FPVA, FPVA-ZLDH (nanocarriers), pristine 5-FU, and FPVAFU-ZLDH (nanoparticles) with normal fibroblast and 3T3 cells. Figure 11 illustrates the percentage cell viability of the 3T3 cells after 3 days incubation for all of the samples. All samples comprising Fe, FPVA, FPVA-ZLDH, 5-FU, and FPVAFU-ZLDH were biocompatible and nontoxic and had a cell viability of 80% after incubation for 3 days at a concentration in the range of 1.25–50 µg. This result indicated the non-toxicity and biocompatibility of the designed nanoparticle formulations and that nanoparticles could target cancer cells without affecting normal cells. No significant difference was observed between sample groups at the concentration in the range of 1.25–50 µg (* *p* value > 0.5).

#### 3.10.2. Anticancer Action against Liver Cancer Cells (HepG2)

The anticancer activity of Fe, FPVA, FPVA-ZLDH (nanocarriers), pristine 5-FU, and FPVAFU-ZLDH (nanoparticles) towards liver cancer cells (HepG2) is shown in Figure 12. The liver cancer cells (HepG2) were incubated with different concentrations of prepared samples for 3 days. The empty nanocarriers Fe, FPVA, and FPVA-ZLDH from concentrations of 1.25–50 µg did not significantly inhibit the activity of liver cancer cells (HepG2). The pristine 5-FU revealed an IC_50_ of 21.54 μg/mL against liver cancer cells (HepG2). The actual amounts of 5-FU in IC50 anticancer nanoparticles (i.e., effective IC_50_) was calculated using HPLC analysis through the percentage of 5-FU drug loading (%) for FPVAFU-ZLDH. The IC_50_ of the FPVAFU-ZLDH nanoparticles was found to be 11.43 μg/mL. These findings propose a more efficient anticancer activity of FPVAFU-ZLDH nanoparticles than a pristine 5-FU drug. A significant difference was observed between FPVAFU-ZLDH nanoparticles and other samples at a concentration in the range of 6.25–100 μg/mL (* *p* < 0.5). A significant difference was found between the 5-FU sample and empty nanocarrier (* *p* < 0.5).

The samples of pristine 5-FU and FPVAFU-ZLDH (nanoparticles) inhibited the activity of the cancer cell line and this effect was dose-dependent. Table 6 shows the IC_50_ values for all of the samples. The calculated values of IC_50_ obtained from the percentage of drug loading indicate more effective anticancer activity of nanoparticles compared to the free form drug.

## 4. Conclusions

One of the most important challenges in nanomedicine is the timely diagnosis and effective treatment of chronic diseases such as cancers. This research aimed to synthesize and characterize magnetic iron oxide nanoparticles with simultaneous applications as a carrier for drug and target delivery. The nanoparticles had a size of 122 nm, with an average diameter of 26 nm. The saturation magnetization value of the nanoparticles was found to be 27 emu/g. The coating agents were found to significantly affect the particle size and the formation of agglomerated structures. The addition of the coating agent resulted in the lowest hydrodynamical particle size (the lowest number of agglomerates) and crystallites, with a particle size distribution that was nearly normal. On the other hand, the addition of coating agents improved the stability and decreased the saturation and magnetic susceptibility of the magnetite nanoparticles. Cell cytotoxicity studies of the nanoparticles towards liver cancer, including the HepG2 cell line and 3T3 cell line, revealed that the synthesized nanoparticles had no cytotoxicity under the tested concentrations. The result of the present study illustrated that the produced nanoparticles have a preliminary essential capability to be used as a carrier for drugs and at the same time, they themselves acted as a target agent.

## Figures and Tables

**Figure 1 polymers-13-00855-f001:**
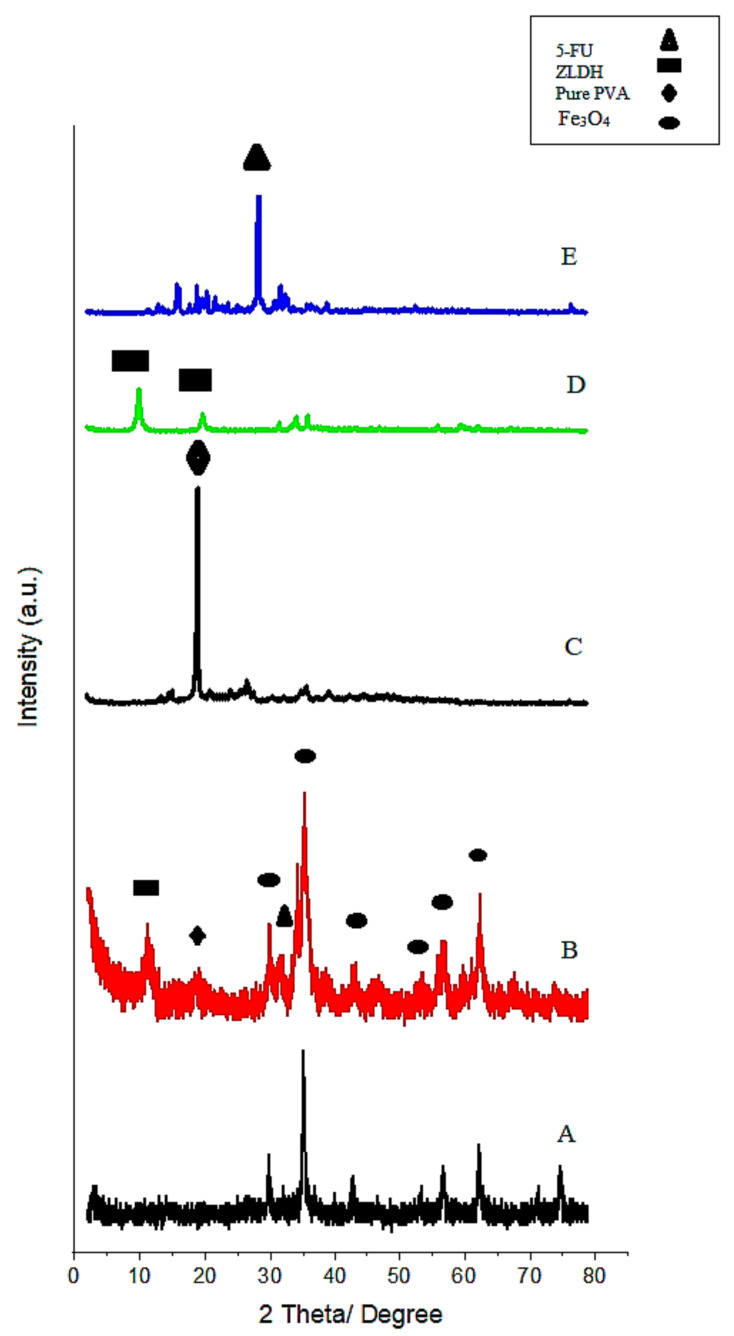
X-ray powder diffraction patterns of the pure Fe_3_O_4_ nanoparticles (**A**) and FPVAFU-ZLDH (**B**), polyvinyl alcohol (PVA) (**C**), ZLDH (**D**), and 5-fluorouracil (5-FU) (**E**).

**Figure 2 polymers-13-00855-f002:**
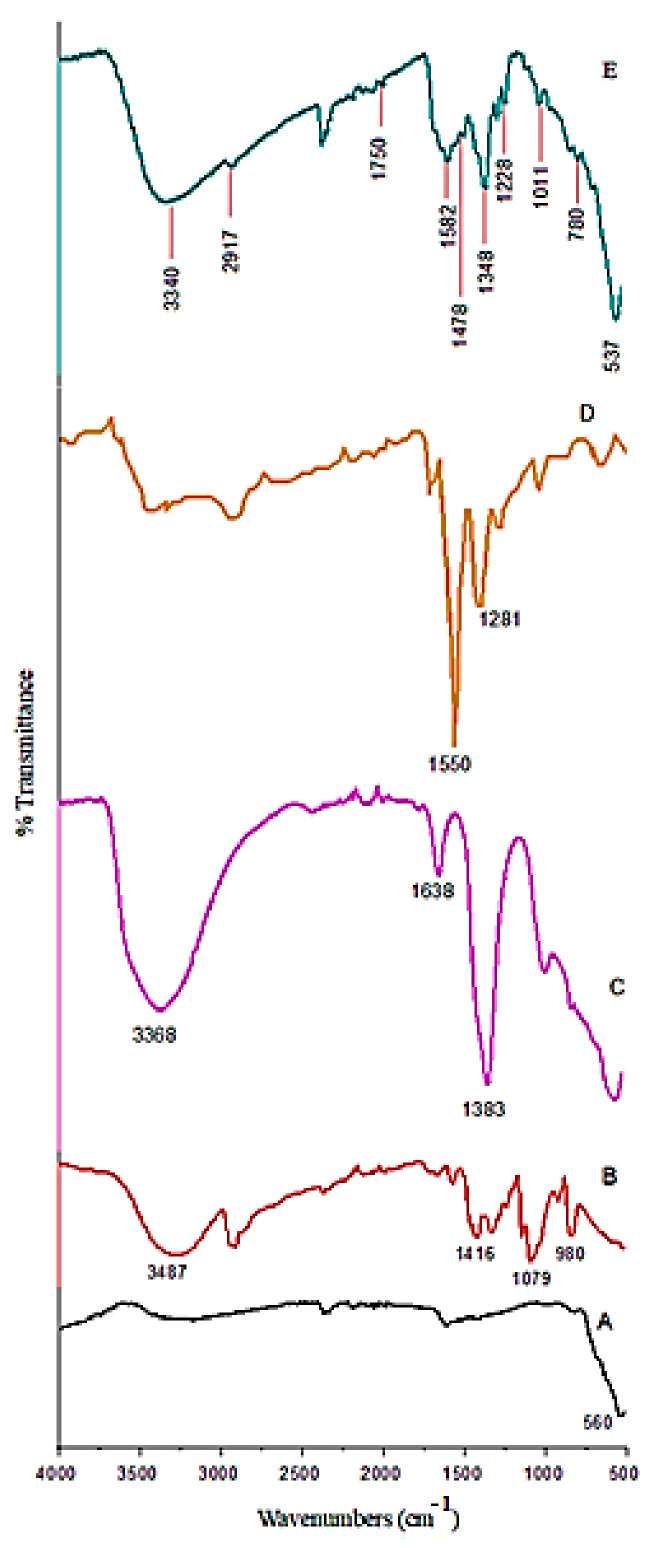
Fourier transformed infrared spectroscopy (FTIR) spectra of the pure Fe_3_O_4_ nanoparticles (**A**), PVA (**B**), Zn/Al-LDH (**C**), 5-FU (**D**), and FPVAFU-ZLDH (**E**).

**Figure 3 polymers-13-00855-f003:**
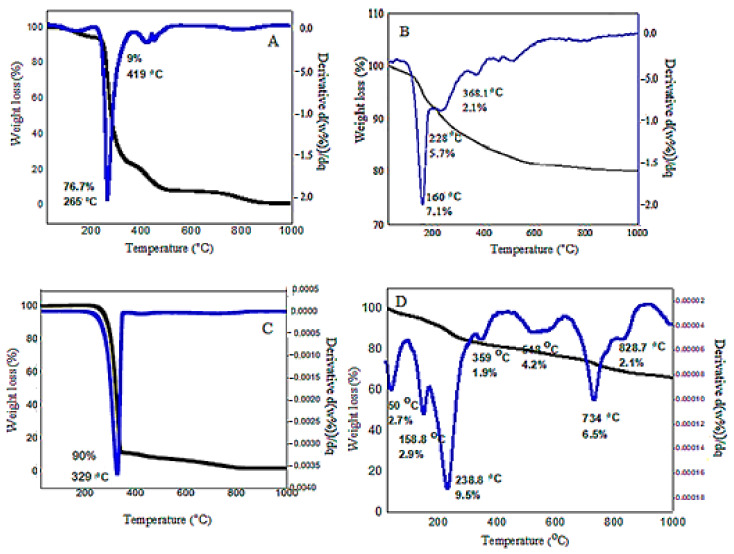
TGA/DTG thermograms for the bare PVA (**A**), ZLDH (**B**), 5-FU (**C**), and FPVAFU-ZLDH (**D**).

**Figure 4 polymers-13-00855-f004:**
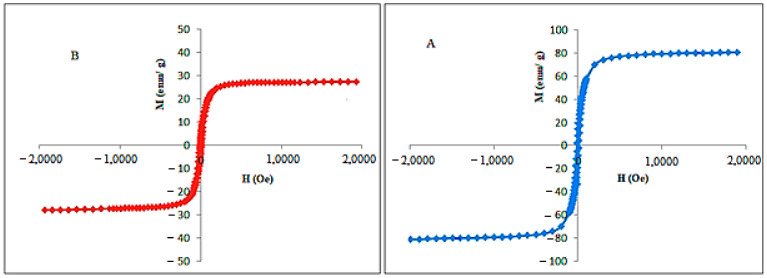
Hysteresis loops for the prepared Fe3O4 nanoparticles (**A**) and FPVAFU-ZLDH (**B**). Notes: The data are presented in terms of Ms, mass magnetization (emu/g), versus H, applied magnetic field (Oe).

**Figure 5 polymers-13-00855-f005:**
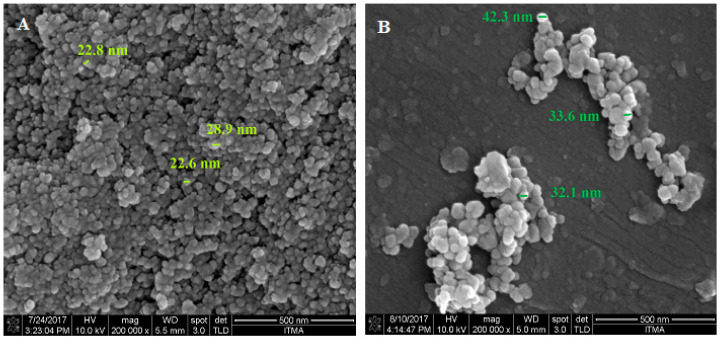
FESEM images of the samples: The pure Fe_3_O_4_ nanoparticles (**A**); FPVAFU-ZLDH (**B**); and EDS data of the prepared samples (**C**). The sample holder is manufactured of aluminum, so the resulting spectrum exhibits a high percentage of aluminum, and it is unreliable to represent the amount of aluminum in the sample.

**Figure 6 polymers-13-00855-f006:**
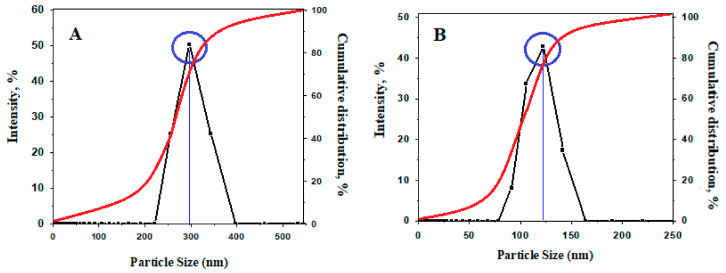
The relative and cumulative particle size distributions of naked Fe_3_O_4_ nanoparticles (**A**) and the FPVAFU-ZLDH (**B**).

**Figure 7 polymers-13-00855-f007:**
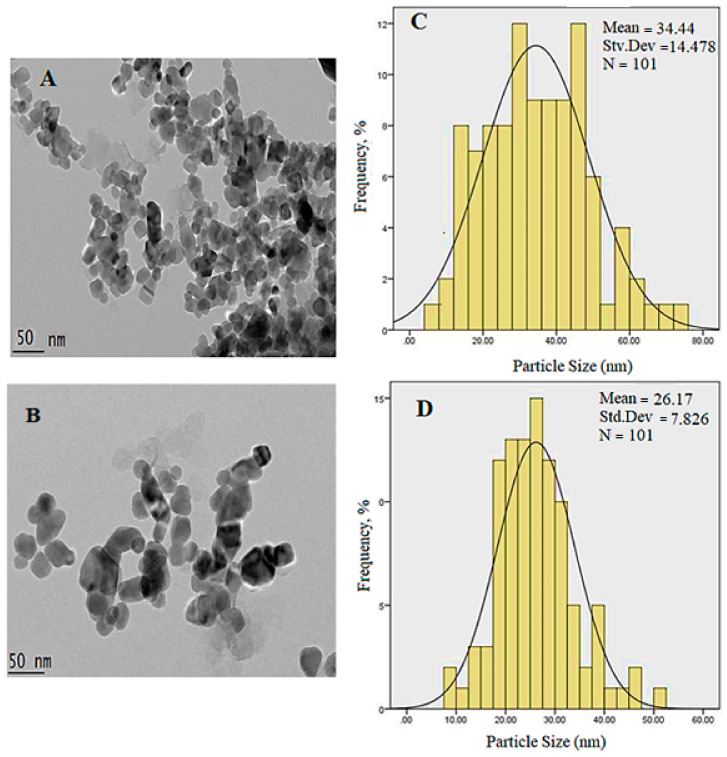
HRTEM micrographs of uncoated and coated samples: (**A**) Uncoated Fe_3_O_4_ nanoparticles (50 nm bar); (**C**) FPVAFU-ZLDH (50 nm bar); and (**B**,**D**) show their particle size distribution, respectively.

**Figure 8 polymers-13-00855-f008:**
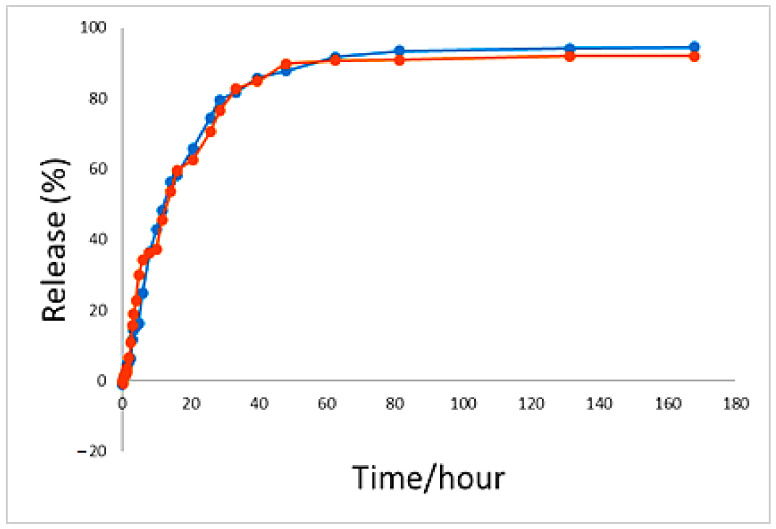
The 5-fluorouracil release profiles from FPVAFU-ZLDH in phosphate-buffered solutions at pH 4.8 and pH 7.4.

**Figure 9 polymers-13-00855-f009:**
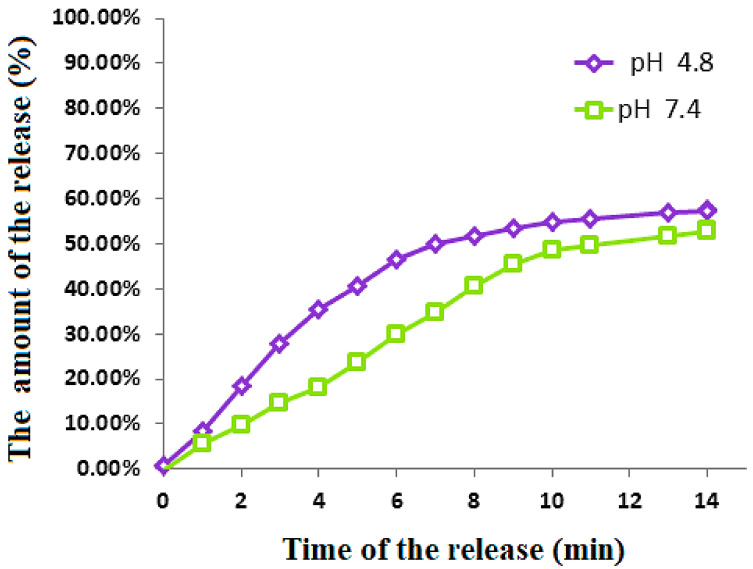
Release of 5- fluorouracil from its physical mixtures and the FPVAFU-ZLDH sample into the phosphate-buffered solutions at pH 4.8 and pH 7.4.

**Figure 10 polymers-13-00855-f010:**
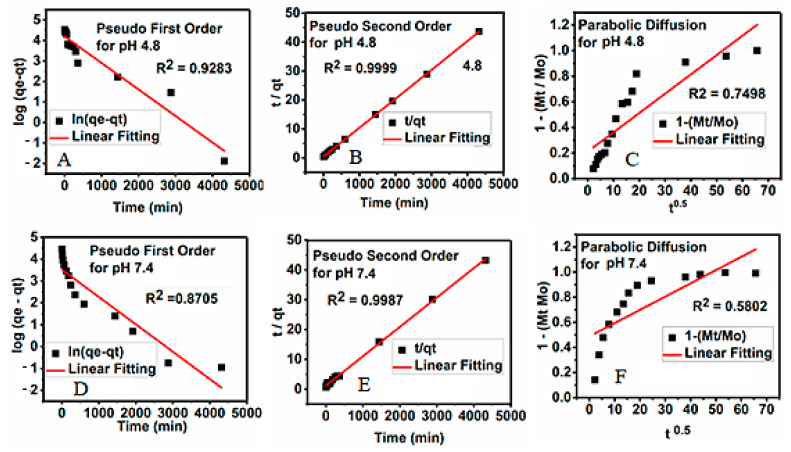
Kinetic parameters for the study of the release of 5-FU from FPVAFU-ZLDH dissolved in dimethyl sulfoxide into different solutions in relation to (**A**) the pseudo-first-order kinetic, (**B**) the pseudo-second-order kinetic, and (**C**) the parabolic diffusion kinetic for pH 4.8, and that dissolved in dimethyl sulfoxide in different solutions in relation to (**D**) the pseudo-first-order kinetic, (**E**) the pseudo-second-order kinetic, and (**F**) the parabolic diffusion kinetic for pH 7.4.

**Figure 11 polymers-13-00855-f011:**
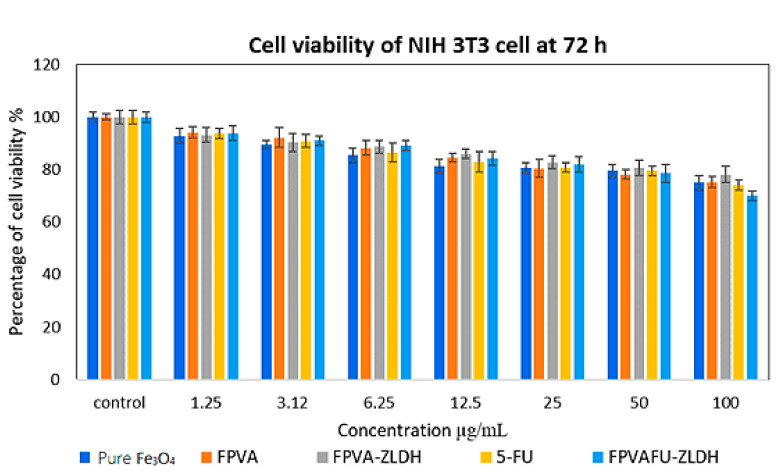
Cytotoxicity assay of pure Fe_3_O_4_ nanoparticles, FPVA, FPVA-ZLDH (nanocarriers), pristine 5-FU, and FPVAFU-ZLDH (nanoparticles) against normal human fibroblast (3T3) cells at 72 h.

**Figure 12 polymers-13-00855-f012:**
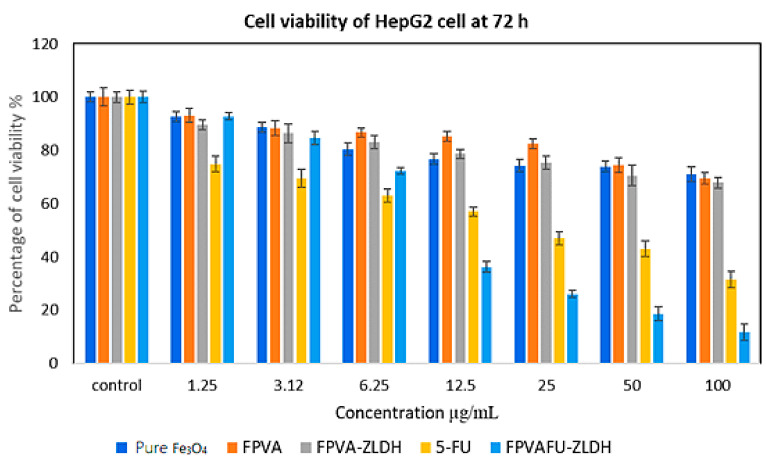
Cytotoxicity assay of pure Fe_3_O_4_ nanoparticles, FPVA, FPVA-ZLDH (nanocarriers), pristine 5-FU, and FPVAFU-ZLDH (nanoparticles) against HepG2 cells at 72 h of incubation.

**Table 1 polymers-13-00855-t001:** Thermal behavior of the nanoparticles.

Samples	T_1_–T_2_ (°C)	T_max_ (°C)	Δm (mg)	Weight Loss (%)
FPVA-FU-ZLDH	33–90	50	0.2	2.7
	102–180	158	0.3	2.9
	180–334	238	1.0	9.5
	334–409	359	0.2	1.9
	459–635	548	0.4	4.2
	635–811	734	0.6	6.5
	811–908	828	0.2	2.1

**Table 2 polymers-13-00855-t002:** Magnetic behavior of the synthesized nanoparticles.

Samples	Ms (emu/g)	Mr (emu/g)	Hci (G)
Fe_3_O_4_	80	1.45	11.5
FPVAFU-ZLDH	27	2.83	7.5

**Table 3 polymers-13-00855-t003:** Elemental analysis of the sample obtained by the ICP-AES and CHNS analyses.

Sample	* C%	* H%	* N%	Zn%	Al%	Fe%
Fe_3_O_4_	0.02	0.54	1.02	-	-	47.00
ZLDH	-	2.37	4.45	6.80	5.20	-
5-FU	52.90	5.01	19.84	-	-	-
PVA	52.05	8.68	1.00	-	-	-
FPVAFU-ZLDH	5.9	1.64	0.32	1.80	2.50	20.80

Noted: Elements with starred were analyzed by CHN analysis and unstarred elements were tested by ICP-OES analysis.

**Table 4 polymers-13-00855-t004:** Percentage of elements of the sample obtained by EDX analyses.

Sample	* C%	* H%	* N%	Zn%	Al%	Fe%
Fe_3_O_4_	-	0.54	0.07	-	-	8.50
ZLDH	-	2.37	0.40	0.10	0.19	-
5-FU	4.40	5.00	1.40	-	-	-
PVA	4.30	8.60	0.07	-	-	-
FPVAFU-ZLDH	0.49	1.60	0.02	0.02	0.10	0.30

Noted: The elements marked with an asterisk (C, H, and N) were examined using the CHNS analysis.

**Table 5 polymers-13-00855-t005:** The correlation coefficient, rate constant, and half-life obtained by fitting the 5-FU release data for the PBS solution at pH 4.8 and pH 7.4.

Sample pH	Saturation Release/%	R^2^			Pseudo Second Order Rate Constant (k(mg/min))	t_1/2_
		**Pseudo-First-Order**	**Pseudo-Second-Order**	**Parabolic- Diffusion**		
4.8	99.99	0.9283	0.9999	0.7498	4.33 × 10^−3^	97
7.4	99.57	0.8705	0.9987	0.5802	4.31 × 10^−3^	84

**Table 6 polymers-13-00855-t006:** The half-maximal inhibitory concentration (IC_50_) value forFe_3_O_4_ nanoparticles, FPVA, FPVA-ZLDH (nanocarriers), pristine 5- fluorouracil, and FPVAFU-ZLDH (nanoparticles) samples tested on 3T3 and HepG2 cell lines.

Nanocomposites IC50 (μg/mL)	3T3 Fibroblast Cells	HepG2 Cells
Pure Fe_3_O_4_ nanoparticles	N.C	N.C
FPVA	N.C	N.C
FPVA-ZLDH	N.C	N.C
5-FU	N.C	21.54
FPVAFU-ZLDH	N.C	11.43

Abbreviation: N.C, no cytotoxicity.

## Data Availability

The raw/processed data required to reproduce these findings cannot be shared at this time as the data also forms part of an ongoing study.

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
