# Peer review of "Dual-Functional Iron Oxide Nanoparticles Coated with Polyvinyl Alcohol/5-Fluorouracil/Zinc-Aluminium-Layered Double Hydroxide for a Simultaneous Drug and Target Delivery System"

_polymers, 2021, doi:10.3390/polym13060855_

Round 1

Reviewer 1 Report

Paper titled (Dual-Functional Iron Oxide Nanoparticles Coated with Polyvinyl Alcohol/5-fluorouracil/Zinc-Aluminium- Layered Double Hydroxide for Simultaneous Drug and Target Delivery System) by Ebadi et al., synthetiized novel nanoparticle formulation containing 5-FU and compared their efficacy to plane 5FU for its cytotoxic activity in HepG2 liver cancer cell line with appropriate control. This is an interesting study and the methodology is supporting the aim of the study.

I have some the following recommendations: 

Introduction : somehow long: the second paragraph can be removed without affecting the integrity of the idea. the other paragraphs can be reduced to make the introduction more concrete.

Methods: the supplier of each chemical (company, town and country) should be provided consistently. 

MWt of the iron chloride (II & III) should be provided

A Shimadzu XRD, PW-6000 Model, Japan : details about the company (town, country) should be provided for this and all similar instruments and software allover the study.

write 5FU in a consistent way (once abbreviated, please use the abbreviation); the same for other abbreviated terms

Figure 8, release % better to be demonstrated per hours on x axis & Y axis: no need for decimal places. 

Any statistical analysis was done to compare the two formulas release at different pH values? why these values were selected and tested?

Any statistical analysis was done for IC50 values in Table 6 and figure 12.

Table 6. title: e half-maximal inhibitory concentration (IC50)! is this a correct definition for IC50? please use the correct term allover the study

In the introduction, authors talked about the toxicity of drugs. Did they measure any toxic manifestation ? kindly comment and discuss the different effects on normal and tumor liver cells.

Author Response

The reply to reviewer comments is attached as a PDF file. 

Reviewer 2 Report

This manuscript is interesting and novel. However, I think that its objective is closer with other MDPI journals such as materials or nanomaterials. I think that this article must be accepted after major revisions, but it should be published in other MDPI journal, being more interesting for their readers. I attached my comments:

1) Abstract must be rewritten to improve its quality and highlight the novelty of this work.

2) Some companies need to indicate the city, state and country.

3) The numbers in Figure 5 are not well visible.

4) Figure 6 must be defined in the text and a leyend must be included.

5) Discussion must be improved.

6) Table 3 and 4: What do the asterisks before carbon, hydrogen, and nitrogen mean?

7) Experimental and results are mixed.

8) Figure 11 and 12: The leyend must be homogenized with the manuscript.

Author Response

(The authors gave the same response as above.)
